# Smart Helmet 5.0 for Industrial Internet of Things Using Artificial Intelligence

**DOI:** 10.3390/s20216241

**Published:** 2020-11-01

**Authors:** Israel Campero-Jurado, Sergio Márquez-Sánchez, Juan Quintanar-Gómez, Sara Rodríguez, Juan M. Corchado

**Affiliations:** 1Laboratoire de l’Informatique du Parallélisme 46 allée d’Italie, 69007 Lyon, France; israel.campero_jurado@ens-lyon.fr; 2BISITE Research Group, University of Salamanca, Calle Espejo s/n. Edificio Multiusos I+D+i, 37007 Salamanca, Spain; srg@usal.es (S.R.); corchado@usal.es (J.M.C.); 3Graduate School in Information Technology and Communications Research Department, Universidad Politécnica de Pachuca, Zempoala Hidalgo 43830, Mexico; juanquintanargomez@micorreo.upp.edu.mx; 4Air Institute, IoT Digital Innovation Hub (Spain), 37188 Salamanca, Spain; 5Department of Electronics, Information and Communication, Faculty of Engineering, Osaka Institute of Technology, Osaka 535-8585, Japan; 6Faculty of Creative Technology & Heritage, Universiti Malaysia Kelantan, Locked Bag 01, 16300 Bachok, Malaysia

**Keywords:** PPE, OHS, risk detection, naive Bayes, support vector machine, convolutional neural network, deep learning, microcontroller

## Abstract

Information and communication technologies (ICTs) have contributed to advances in Occupational Health and Safety, improving the security of workers. The use of Personal Protective Equipment (PPE) based on ICTs reduces the risk of accidents in the workplace, thanks to the capacity of the equipment to make decisions on the basis of environmental factors. Paradigms such as the Industrial Internet of Things (IIoT) and Artificial Intelligence (AI) make it possible to generate PPE models feasibly and create devices with more advanced characteristics such as monitoring, sensing the environment and risk detection between others. The working environment is monitored continuously by these models and they notify the employees and their supervisors of any anomalies and threats. This paper presents a smart helmet prototype that monitors the conditions in the workers’ environment and performs a near real-time evaluation of risks. The data collected by sensors is sent to an AI-driven platform for analysis. The training dataset consisted of 11,755 samples and 12 different scenarios. As part of this research, a comparative study of the state-of-the-art models of supervised learning is carried out. Moreover, the use of a Deep Convolutional Neural Network (ConvNet/CNN) is proposed for the detection of possible occupational risks. The data are processed to make them suitable for the CNN and the results are compared against a Static Neural Network (NN), Naive Bayes Classifier (NB) and Support Vector Machine (SVM), where the CNN had an accuracy of 92.05% in cross-validation.

## 1. Introduction

Industrial security is achieved when adequate measures and procedures are applied to obtain access to, handle or generate classified information during the execution of a classified contract or program. Industrial safety is the set of rules and activities aimed at preventing and limiting the potential risks associated with an industry, including both transient and permanent risks [1,2].

Many safety protocols have been proposed to improve the quality of life of workers using different techniques [3,4]. Several studies have examined how the availability of artificial intelligence (AI) techniques could affect the industrial organization of both AI service providers and industries adopting AI technology [5]. Above all, the impact of AI on industry 4.0 and its possible applications in other fields have been studied in depth [6].

In recent years, research has also been conducted on the applications of AI in the manufacturing industry [7,8,9,10,11]. The system architecture described in the article integrates technology together with communication systems and permits analyzing intelligent manufacturing. The provided information shows an overview of the possible applications of AI in all industrial areas.

AI allows to maximize decision making in simple or very complex situations. The AI boom that has taken place in the last decades has led to the development of countless AI applications in numerous areas. At present, increasingly better solutions are available to protect the lives of workers when they are exposed to high-risk conditions. That is why, in industry, AI is combined with security measures in order to create an environment that offers better conditions for industrial development.

The objective of the proposed device is to improve occupational health and safety (OHS); increasing employee performance by reducing the probability of illness, injury, absence or death [12]. Another objective is to contribute to the third wave, as proposed by Niu et al. [13,14], through the implementation of intelligent systems for early risk detection in the working environment.

Different studies have been conducted in creation of devices for occupational safety and health (OSH), which indicate the need to implement increasingly innovative solutions for workers in high-risk areas. For example, in 2014 [15] a study was conducted among 209 welders in India and it was found that all of them had more than 2 injuries and 44% (92) of them had more than 10 injuries. Furthermore, in 2020 [16] an analysis of workplace-related injuries in major industries such as agriculture, construction, manufacturing and health care has been carried out. The data for this analysis have been obtained from a Bureau of Labor Statistics and it was found that from 1992 to 2018, the number amounted to 4,471,340 injuries in the upper extremities, 3,296,547 in the lower extremities, and 5,889,940 in the trunk (*p* < 0.05). Therefore, the motivation behind this research is to propose an innovative helmet with different sensors such as temperature, humidity and atmospheric pressure, the force exerted between the helmet and the head of the user, the variations in axes, air quality and luminosity, through specialized IoT modules being able to have a faster reaction time to an accident in a work team. All the research papers that address the problem of occupational safety and health (OSH) are summarised in Table 1 with the purpose of comparing the improvements and advantages of similar research.

The information coming from the sensors is analyzed through a platform known as ThingsBoard. Independent alarms are configured using this information. Likewise, the data coming from the sensors are adapted to classify them in a Convolutional Neural Network, whose accuracy is of 92.05% in cross-validation compared to 3 other supervised learning models.

The remaining part of this work is organized as follows: Section 2 gives an overview of the related literature. Section 3 describes the system design. A multisensory helmet with communication in IIoT and AI-based information analysis is presented in Section 4. Finally, the last section describes future lines of research.

## 2. Related Works

Protective equipment is of obligatory use in cases where the safety of the worker is at risk. However, detecting hazardous situations in a timely manner is not always possible, leading to the occurrence of accidents. Such events call the worker’s health and safety into question; the confidence of the worker in the company for which they work decreases [17,18,19]. For effective prevention of injuries or fatal accidents in the working environment, the integration of electronic components is crucial given their ability for early risk detection. The research of Henley, E.J. and Kumamoto, H [20] proposed a quantitative approach for the optimal design of safety systems which focused on information links (human and computer), sensors, and control systems. In 2003, Condition Monitoring (CM) was addressed in the research of Y. Han and Y. H. Song [21] including a review of popular CM methods, as well as the research status of CM transformer, generator, and induction motor, respectively. In December 2001, the factor structure of a safety climate within a road construction organization was determined by A.I Glendon and D.K Litherland [22]; a modified version of the safety climate questionnaire (SCQ). They also investigated the relationship between safety climate and safety performance. In March 2011, Intelligent Internet of Things for Equipment Maintenance (IITEM) was presented by Xu Xiaoli et al. [23]. The static and dynamic information on electrical and mechanical equipment is collected by IITEM from all kinds of sensors, and the different types of information are standardized, facilitating Internet of Things information transmission [24,25]. The investigations that address motion monitoring and sensor networks have been compiled in Table 2.

Moreover, an Accident Root Causes Tracing Model (ARCTM), tailored to the needs of the construction industry, has been presented by Tariq S. Abdelhamid and John G. Everett [26]. In January 2010, guidelines for identifying, analyzing and disseminating information on near misses at construction sites were defined by Fabricio BorgesCambraia et al. [27]. In September 2013, three case studies were presented by Tao Cheng and Jochen Teizer [28] which employed methods for recording data and visualizing information on construction activities at a (1) simulated virtual construction site, (2) outdoor construction setting, and (3) worker training environment. Furthermore, systems theory has been incorporated in Heinrich’s domino theory by Seokho Chia and Sangwon Han [29] to explore the interrelationships between risks and to break the chain of accident causation. In April 2008, the reasons for which construction workers engage in unsafe behavior were discussed in the empirical research of Rafiq M. Choudhry and Dongping Fang [30]. Interviews were conducted in Hong Kong with workers who had been accident victims. In addition, Daniel Fitton et al. [31] applied augmented technology with sensing and communication technologies which can measure use in order to enable new pay-per-use payment models for equipment hire. The areas in which it is necessary to create a safer working environment are listed in Table 3. This can be achieved through the use of sensors for the monitoring environmental parameters and capturing motion.

In December 2008, the underlying biomechanical elements required to understand and study human movement were identified by A. Godfrey et al. [32]. A method for investigating the kinematics and dynamics of locomotion without any laboratory-related limitations has been developed by Yasuaki Ohtaki et al. [33]. In April 2012, the usage of the Unscented Kalman Filter (UKF) as the integration algorithm for the inertial measurements was proposed by Francisco Zampella et al. [34]. Furthermore, in 2012, a micro wearable device based on a tri-axis accelerometer was introduced by Yinghui Zhou et al. [35]. It can detect change in the acceleration of the human body on the basis of the position of the device. In 2009, a method for the recognition of daily human activities was developed by Chun Zhu and Weihua Sheng [36]. This method involved fusing the data from two wearable inertial sensors attached to the foot and the waist of the subject. In October 2012, Martin J.-D. Otis and Bob-Antoine J. Menelas [37] reported an ongoing project whose objective was to create intelligent clothes for fall prevention in the work environment. In 2007, a signal transform method, called Common Spatial Pattern, was introduced by Hong Yu et al. [38] for Electroencephalographic (EEG) data processing. In March 2006, the development history of a wearable, called the scalable vibrotactile stimulus delivery system, was presented by Robert W. Lindeman et al. [39]. In 2014, an objective and real-time approach based on EEG spectral analysis for the evaluation of fatigue in SSVEP-based BCIs was proposed by Teng Cao et al. [40].

Thanks to the implementation of communication technologies, it is possible to notify both the managing staff and the workers about the hazards encountered in a particular working area. A helmet that implements Zigbee transmission technologies for the analysis of variables such as humidity, temperature and methane in mines has been developed by Qiang et al. (2009) [41]. This helmet helps decrease the risk of suffering an accident during the coal extraction process. An intelligent helmet for the detection of anomalies in mining environments was also proposed by Nithya et al. (2018) [42]. This research points to the possibility of integrating components in the PPE that would alert the worker of the presence of danger. Moreover, the vital signs of the worker are monitored by their helmet, making it possible to monitor their state of health. An emergency button on the helmet is used for the transmission of alerts via Zigbee technologies to the personnel nearest to the working environment. Accelerometers have been integrated in safety helmets by Kim et al. (2018) [43], with the purpose of detecting if the safety helmet is being worn properly, improperly or not worn at all while the worker performs their tasks. In December 2016, a framework for real-time pro-active safety assistance was developed by Yihai Fang et al. [44] for mobile crane lifting operations.

Ensuring the physical well-being of workers is the responsibility of employers. Better protection is offered to today’s workers thanks to PPE helmets by protecting the worker from blows to the head. However, monitoring other aspects for the worker’s security is important in some cases. Li et al. (2014) [45] developed a helmet which, by means of sensors, measures the impact of blows to the worker’s head. Sensors for brain activity detection are also implemented in the helmet. In terms of movement, identifying the position of the worker is essential in order to detect falls that result in physical injury or fatal accidents.

In 2019, Machine Learning (ML) algorithms for the prediction and classification of motorcycle crash severity were employed in a research by Wahab, L., and Jiang, H. [46]. Machine-learning-based techniques are non-parametric models without any presumption of the relationships between endogenous and exogenous variables. Another objective of this paper was to evaluate and compare different approaches to modeling motorcycle crash severity as well as investigating the risk factors involved and the effects of motorcycle crashes. In 2015, a scalable concept and an integrated system demonstrator was designed by Bleser, G. et al. [47]. The basic idea is to learn workflows from observing multiple expert operators and then transferring the learned workflow models to demonstrate the severity of motorcycle crashes. In 2019, an intelligent video surveillance system which detected motorcycles automatically was developed by Yogameena, B., Menaka, K., and Perumaal, S. S. [48]. Its purpose was to identify whether motorcyclists were wearing safety helmets or not. If the motorcyclists were found without the helmet, their License Plate (LP) number was recognised and legal action was taken against them by the traffic police and the legal authority, such as assigning penalty points on the motorcyclists’ vehicle license and Aadhar Number (Applicable to Indian Scenario). In 2017, a comparison of four statistical and ML methods was presented by Iranitalab, A., and Khattak [49], including Multinomial Logit (MNL), Nearest Neighbor Classification (NNC), Support Vector Machines (SVM) and Random Forests (RF), in relation to their ability to predict traffic crash severity. A crash costs-based approach was developed to compare crash severity prediction methods, and to investigate the effects of data clustering methods—K-means Clustering (KC) and Latent Class Clustering (LCC)—on the performance of crash severity prediction models. These novel proposals are compiled in Table 4. They employ artificial intelligence and machine learning, and suppose a significant improvement in different scenarios.

In 2005, the results obtained with the random forest classifier were presented in the research of M. Pal [50] and its performance was compared with that of the support vector machines (SVMs) in terms of classification accuracy, training time and user defined parameters. In January 2012, the performance of the RF classifier for land cover classification of a complex area was explored by V. F. Rodriguez-Galiano et al. [51]; the evaluation was based on several criteria: mapping accuracy, sensitivity to data set size and noise. Furthermore, in February 2014, a random forest classifier (RF) approach was proposed by Ahmad Taher Aza et al. [52] for the diagnosis of lymph diseases. In April 2016, the use of the RF classifier in remote sensing was reviewed by Mariana Belgiua and Lucian Drăguţ [53]. Besides, in 2015, machine learning approaches including k-nearest neighbor (k-NN), a rules-based classifier (JRip), and random forest, were investigated by Esrafil Jedari et al. [54] to estimate the indoor location of a user or an object using RSSI based fingerprinting method. Finally, in July 2011, a method utilizing Healthcare Cost and Utilization Project (HCUP) dataset was presented by Mohammed Khalilia et al. [55] for predicting disease risk in individuals on the basis of their medical history.

With regard to CNN in 2020, an automated system for the identification of motorcyclists without helmets from real-time traffic surveillance videos was presented by Shine L. and Jiji, C. V. [56]. A two-stage sorter was used to detect motorcycles in surveillance videos. The detected motorcycles were fed in a helmet identification stage based on a CNN. Moreover, in July 2019, the same approach to detecting the absence of helmets on motorcyclists with or without helmets was presented by Yogameena B. et al. [48]; it was different in that it combined a CNN with a Gaussian Mixture Model (GMM) [57]. Furthermore, in 2020, a system that uses image processing and CNN networks was developed by Raj K. C. et al. [58] for the identification of the motorcyclists who violate helmet laws. The system includes motorcycle detection, helmet vs. helmetless classification and motorcycle license plate recognition. As can be observed, CNNs have been used mainly for real-time image processing. However, the use of CNN for linear data evaluation is proposed in this paper. Here, CNN is integrated (input–output) in a rules model for the classification of different problems in working environments. The presented papers are examples and inspired the given research as a support for this paper. A diagram of the most represented technologies in the state of the art is given in Figure 1. These technologies are the main basis of the proposal.

## 3. Smart Helmet 5.0 Platform

There are different methodologies for carrying out research on electronics and system design. Thus, in this section, a description of the hardware and software used for the development of the fifth version of the smart helmet will be presented, and the procedure followed for its subsequent validation through the AI model will be detailed. The four previous helmets included less sensorisation and conectivity, which is why we developed a new version with all the improvements.

### 3.1. Hardware Platform

The structure followed in the development of the proposed helmet are the steps involved in the prototype development methodology, identifying the parameters to be monitored in the environment. A Job Safety Analysis (JSA) was performed, identifying the risk factors that lead to injuries and accidents in the worker [59]. The deficiencies that have been observed are presented in Table 5 for work places such us mines, construction places and electrical work areas. They are related to aspects such as lighting, detection of blows to the worker’s helmet (PPE detection), dangerous temperature levels for human activityand poor air quality in the environment. Other parameters that could be interesting such as, noise, rate pulse and body temperature are implemented in other devices for better ergonomics.

Given the above, a series of specialized sensors are proposed to counteract the difficulties that usually occur in a high-risk work environment [1], see Table 6. As seen in the literature review, agriculture and industrial activities involve high risk, among others.

In terms of the transmission of information from sensors, the use of Wi-Fi technologies has been selected due to their ability to transmit the information in Local Area Networks (LAN) to a web server responsible for collecting, processing and transmitting anomaly warnings to the worker or administrative personnel. The following describes the system design and the interaction of the components.

The elements used in the smart helmet and the risks it seeks to prevent or detect are detailed below. The operation of the Smart PPE and the distribution of the circuits will also be discussed. In addition, the architecture and technologies are explained, as well as the operating rules of the different sensors and actuators that make up the system. Finally, their communication system is considered, as well as the technology used for both the management of the data and for its visualization and treatment once obtained.

The aim of this Smart PPE is to protect the operator from possible impacts, while monitoring variables in their environment such as the amount of light, humidity, temperature, atmospheric pressure, presence of gases and air quality. At the same time, the Smart PPE is to be bright enough to be seen by other workers, and the light source will provide extra vision to the operator. All these alerts will be transmitted to the operator by means of sound beeps. The sensors described below were selected as part of the set of electronic devices to be implemented:Temperature, gas and pressure sensor;Brightness sensor;Shock sensor;Accelerometer.

In the process of the visualization of environmental data, a LED strip is deployed on the helmet as a means of notifying the worker of anomalies through color codes presented in the environment. The block diagram shown in Figure 2 is a representation of the electronic system integrated in the helmet.

The specifications of the sensors and the microcontroller used to monitor the environment are defined as follows:

The component used to supervise the parameters of gas, pressure, temperature and humidity is the low power environmental sensor DFRobot BME680. It is a MEMS (Micro-Electromechanical System) multifunctional 4 in 1 environmental sensor that integrates a VOC (Volatile Organic Compounds) sensor, temperature sensor, humidity sensor and barometer. The environmental pressure is subject to many short-term changes caused by external disturbances. To suppress disturbances in the output data without causing additional interface traffic and processor work load, the BME680 features an internal IIR filter. The output of the subsequent measurement step is filtered using the following Equation (Equation 1):(1)$$\begin{array}{c} x_{filt}\left[n\right]=\frac{x_{filt}\left[n-1\right]*\left(c-1\right)+x_{ADC}}{c} \end{array}$$
where $x_{filt}\left[n-1\right]$ is the data coming from the current filter memory, and $x_{ADC}$ the data coming from current ADC acquisition and where $x_{filt}\left[n\right]$ denotes the new value of filter memory and the value that will be sent to the output registers.

The sensor implemented for the monitoring of the level of brightness is the ALS-PT19 ambient light sensor. Due to the high rejection ratio of infrared radiation, the spectral response of the ambient light sensor resembles that of the human eyes.

The sensor implemented for shock detection is a sensitive force resistor, the sensor emits shock alerts if the readings obtained in the environment exceed a threshold value.

The sensor responsible for the detection of falls suffered by the worker is the MPU6050 module, it is an electronic component that has six axes (three corresponding to the gyroscope system and three to the accelerometer) making it possible to obtain the values of positioning in the X, Y and Z axes.

The light source integrated in the helmet is a NeoPixel Adafruit LED strip, the component integrates a multicolor LED in each section of the strip. The algorithm implemented in the microcontroller is configured in such a way that it is possible to control the color of the LED strip.

The microcontroller used for processing, transmitting and displaying the information transmitted to the web platform is the dual-core ESP-WROOM-32 module of the DFRobot FireBeetle series, which supports communication through Wi-Fi and Bluetooth. The main controller supports two power methods: USB and 3.7 V external lithium battery.

The components are integrated in the microcontroller, which obtains and processes the information coming from the sensors. This information is then transmitted to the implemented web server by means of the Wi-Fi module. The designed electronic system is located in the backside of the helmet, as shown in Figure 3. It also integrates a lamp which is activated automatically if the brightness value of the sensor is below the threshold value established in the programming of the microcontroller. The information transmitted by the helmet can be viewed on a web platform.

This section describes the developed software and the interaction that takes place between the different components.

### 3.2. Intelligence Module

Firstly, the communication between the active sensors is enabled by Thingsboard. ThingsBoard is an open source IoT platform for data collection, processing, visualization and IoT device management. It is free for both personal and commercial use and can be implemented anywhere.

It enables device connectivity through industry-standard IoT protocols (MQTT, CoAP and HTTP) and supports both cloud and on-premise deployments. ThingsBoard combines scalability, fault tolerance and performance, ensuring that the users’ data are never lost.

ESP32 is a series of low-power, low-consumption system-on-a-chip microcontrollers with integrated Wi-Fi and dual-mode Bluetooth, as mentioned in the previous section. The device is responsible for transmitting the information to the ThingsBoard platform and its subsequent processing by the intelligent model see Figure 4, to interact with the helmet.

Simple steps are required to link the devices to the platform:The automatically generated access token is copied from the Access token field.Go to Devices, locate ESP32 device, open the device details, and select the Latest telemetry tab.It is now possible to view the data regarding an asset.

The data obtained through ThingsBoard is later processed by an intelligent model, the model confirms or denies the existence of a real emergency. This is the reason why configuring the platform is very important.

An association must be created between the different values of the sensors and the corresponding response. Once these associations are created, it is possible to modify any value depending on the values to be tested empirically or in the alarms. Alarms are configured in the device settings so that the respective notifications appear on the panel. A rule chain must be added.

A selection of the attributes placed on the server and on the device’s threshold panel must be carried out. The names of the attributes on the server must correspond with those on the panel so that when the data are dynamically configured, they will be recognized correctly and will appear on the diagram generated by the platform, Figure 5.

Subsequently, in the script block, it is verified that the information coming from the device does not exceed the established threshold value. If the script is positive, an alarm is configured and the information to be displayed is defined.

Moreover, the root string, which is in charge of obtaining and processing the information coming from the devices, has been modified. In this case, an originator type section has been added, where the devices that transmit the information are identified. Likewise, code strings have been generated to implement the customized code blocks in the panels. Finally, the information on the data panel may be visualized.

In cases where it is not necessary to perform this procedure, it is possible to view the notifications generated by the different devices. To this end, it is necessary to enter the Device section. Select one of the devices for which an alarm has been configured and go to the alarms tab, see Figure 6, where the notifications generated by that device are displayed.

Once the alarms have been configured on the platform, validation is carried out through the explanation of the AI [60].

## 4. Platform Evaluation

This section compares the different algorithms used in the state of the art to solve problems similar or related to the one being addressed here [61,62,63,64], these models have been accepted for real world problems due to their dataset results with data unbalance and saturation issues, this comparison will be performed with the same amount of data and on an objective quantitative basis. Furthermore, the present proposal is described in detail.

### 4.1. Data Model

In this study, samples of data from a real environment have been obtained, where a subject was subjected to various scenarios in simulated environments, considering the different risks that could arise. The five analyzed parameters are shown in Table 7. The acquired dataset consists of a total of 11,755 samples, where five descriptive variants are proposed with respect to the target of the study.

This research tackles a multi-class type of problem, for this reason there is a set of labels that have a different meaning. When the programming of the microcontroller was carried out, the different parameter values that could trigger an alarm signal were investigated, for example, if the air quality falls below the threshold (measured by the Air Quality Index, AQI) it is possible to associate this situation with the values for other parameters measured by neighboring sensors. The 12 labels proposed in this work are described below, where research was carried out on the most common problems in industrial areas and from there the type of sensors in the helmet were included [65,66]:0.Good for health air (AQI from 0 to 50) with sufficient illumination in the working environment.1.Moderate air quality (AQI of 51 to 100) with slight variation in temperature and humidity.2.Harmful air to health for sensitive groups (AQI 101-150) with moderate variation in temperature and humidity.3.Harmful air to health (AQI 151 to 200) with considerable variation in temperature and humidity4.Very harmful air to health (AQI 201 to 300) with high variation in temperature and humidity.5.Hazardous air (AQI greater than 300) with atypical variation in temperature and humidity.6.Lack of illumination and variation equivalent to a fall in axes.7.Lack of illumination and variation equivalent to a fall in axes and considerable force exerted on the helmet.8.Atypical variation on the detected axes and moderate force detected on the FSR.9.Illumination problems, air quality and sudden variation in axes.10.Very high force exerted on the FSR.11.Variation in axes with illumination problems.12.Outliers on the 5 sensors.

Once the information has been understood, it is cleaned. As proposed by [67], the data were cleaned due to common problems such as missing values solved with the clamp transformation, see Equation (Equation 2).
(2)$$a_{i}=\left\{\begin{array}{cc} lower & ifa_{i}<lower\\ upper & ifa_{i}>upper\\ a_{i} & Otherside \end{array}\right\}$$
where $a_{i}$ represents the i-th sample of the data set, lower and upper thresholds respectively.

The upper and lower thresholds can be calculated from the data. A common way of calculating thresholds for the clamp transformation is to establish:The lower threshold value $=Q_{1}-1.5IQR$;The upper threshold value $=Q_{3}+1.5IQR$.

Where $Q_{1}$ is the first quartile, $Q_{3}$ is the third quartile and IQR is the interquartile range ($IQR=Q_{3}-Q1$ the interquartile range). Any value outside these thresholds would become the threshold values. This research takes into account the fact that the variation in a data set may be different on either side of a central trend. Each sample that had missing data was eliminated so as not to bias the model. However, the search for outliers was only used to find erroneous data generated by the electronic acquisition system since outliers usually provide a large source of information for the analysis of a dataset.

### 4.2. Intelligent Models Evaluation

The comparison part describes each of the models used for the current project, detailing the Support Vector Machine, Naïve Bayes classifier, Static Neural Network and a Convolutional Neural Network. Each model used the dataset of 11,755 where 80% was used for modeling and 20% for evaluation, in other words 9404 in training and 2351 in evaluation. The following confusion matrices reports the result of the validation and after that, we include a figure for each model in order to present the information clearly. It is worth mentioning that all models were trained and validated with the same data division in relation 80-20, it is also notable above an imbalance of classes, given the imbalance some models had unfavorable behavior in cross validation. To handle the imbalance it is possible to opt for techniques such as oversampling or undersampling but it is not desired to change the quality of the data, that is why the model with the best performance will be chosen and evaluated with 10 folds for validation.

#### **Support** **Vector** **Machine**

SVMs belong to the categories of linear classifiers, since they introduce linear separators better known as hyperplanes, regularly made within a space transformed to the original space.

The first implementation used for the multi-class classification of SVM is probably the one against all method (one-against-all). The SVM is trained with all the examples of the m-th class with positive labels, and all the other examples with negative labels. Therefore, given the data of the $\left(x_{1},y_{1}\right),\ldots ,\left(x_{l},y_{l}\right)$ where $x_{i}\in R^{n},i=1,\ldots ,l$ y $y_{i}\in \left\{1,\ldots ,k\right\}$ is the class of $x_{i}$, the m-th SVM, and solve the problem in Equation (Equation 3) [68], which involves finding a hyper plane so that points of the same kind are on the same side of the hyperplane, this is finding a *b* and *w* such:(3)$$\begin{array}{c} y_{i}\left(w^{\prime }x_{i}+b\right)>0,\;i=1,\ldots ,N \end{array}$$

Equation (Equation 4) looks for a hyper plane to ensure that the data are linearly separable.
(4)$$\begin{array}{c} \underset{1\leq i\leq N}{min}\;\;\;y_{i}\left(w^{\prime }x_{i}+b\right)\geq 1 \end{array}$$
where $w\in R^{d},\;b\in R$ and the training dataset $x_{i}$ is mapped to a higher dimensional space. Thus, it is possible to search among the various hyperplanes that exist for the one whose distance to the nearest point is the maximum, in other words, the optimum hyperplane [68], see Equation (Equation 5).
(5)$$\begin{array}{c} \underset{w,b}{min}\frac{1}{2}w^{T}w\\ individual\;\;a\;\;y_{i}\left(w^{T},x_{i}+b\right)\geq 1,\forall_{i} \end{array}$$

Given the above, we are looking for a plane with the maximum distance between the samples of different classes on a higher dimension. As mentioned above, the SVM was of the type one against all in the mathematical description since it is a multi-class problem. Furthermore, the type of kernel was linear. The modeling was performed and the confusion matrix was obtained with 20% of data for evaluation. The accuracy of each class in comparison to the others can be observed in Table 8. The SVM was the model with the worst performance out of the four evaluated according to the recommendation in the literature where the overall accuracy was 68.51%.

#### **Naive** **Bayes** **Classifier**

A Gaussian NB classifier is proposed that is capable of predicting when an accident has occurred in a work environment through different descriptive characteristics, which is based on Bayes’ theorem. Bayes’ theorem establishes the following relationship, given the class variable *y* and the vector of the dependent characteristic $x_{1}$ through $x_{n}$ [69,70], Equation (Equation 6).
(6)$$P\left(y|x_{1},\ldots ,x_{n}\right)=\frac{P\left(y\right)P\left(x_{1},\ldots x_{n}|y\right)}{P(x_{1},\ldots ,x_{n})}$$
where ∀i, the relationship can be simplified as shown in Equation (Equation 7).
(7)$$P\left(y|x_{1},\ldots ,x_{n}\right)=\frac{P\left(y\right)\prod_{i=1}^{n}P\left(x_{i}|y\right)}{P(x_{1},\ldots ,x_{n})}$$
where $P(x_{1},\ldots ,x_{n})$ is constant based on the input; the classification rule presented in Equation (Equation 8) can also be used.
(8)$$\begin{array}{c} P\left(y|x_{1},\ldots ,x_{n}\right)\propto P\left(y\right)\prod_{i=1}^{n}P\left(x_{i}|y\right)\\ ⇓\\ \hat{y}=\arg\max_{y}P\left(y\right)\prod_{i=1}^{n}P\left(x_{i}|y\right), \end{array}$$

The difference in the distributions of each class in the dataset means that each distribution can be independently estimated as a one-dimensional distribution. This in turn helps reduce the problems associated with high dimensionality. For a Gaussian NB classifier the probability of the characteristics is assumed to be Gaussian, see Equation (Equation 9).
(9)$$P\left(x_{i}|y\right)=\frac{1}{\sqrt{2\pi \sigma_{y}^{2}}}\exp\left(-\frac{{\left(x_{i}-\mu_{y}\right)}^{2}}{2\sigma_{y}^{2}}\right)$$

In other words, in order to use the NB classifier in the grouping of the different work circumstances that put the worker at risk, it is assumed that the presence or absence of a particular characteristic is not related to the presence or absence of any other characteristic, given the variable class. The confusion matrix of the NB is shown in Table 9, where on average the accuracy was of 78.26%.

#### **Static** **Neural** **Network**

Neural networks are simple models of the functioning of the nervous system. The basic units are the neurons, which are usually organized in layers. The processing units are also organized in layers. A neural network normally consists of three parts [71]:An input layer, with units representing the input in the dataset.One or more hidden layers.An output layer, with a unit or units representing the target field or fields.

The units are regularly connected with varying connection forces (or weights). Input data is presented in the first layer, and values are propagated from one neuron to another in the next layer. At the end, a result is sent from the output layer. All the weights assigned to each layer are random in the first instance of the training. However, there are a series of methods that can be employed to optimize this phase. Furthermore, the responses that result from the network are offline. The network learns through training [71]. Data for which the result is known are continuously presented to the network, and the responses it provides are compared with the known results.

The use of a static NN is proposed in this research. The performance of the classic model Adam has been compared with the performance of a CNN. The architecture of the NN is shown in Figure 7, which is a three-layer static model, where the first layer contains five neurons that correspond to each of the five data being obtained from the multisensory case, the hidden layer has 32 neurons with the ReLU activation function and finally the output layer has 12 neurons representing the situations a worker may find themselves in. They range from safe to risky situations. The last layer has a SoftMax activation function because it is a multi-class problem. The learning step was 0.05 and the model was trained with 500 epochs. In which the approach for the proposed structure is based on “trial and error”, since as it is well known establishing a neural network is more an art than a science. That is why the number of neurons on the second layer was modified, which obtained a better result than adding other layers on the network. However, CNN showed better results than the rest of the models with a predetermined structure (12 neurons in the hidden layer).

The result of the static NN are given in Table 10. It is possible to observe its performance was not very different from the NB classifier, where an average accuracy of 78.56% was obtained.

### 4.3. Convolutional Neural Network

A Convolutional Neural Network (CNN) was the selected model, it is a deep learning algorithm mainly used to work with images in which it is possible to use an input image (instead of a single vector as in static NNs), assign importance, weights, learnable biases to various aspects/objects of the image and be able to differentiate one from another [72]. The advantage of NNs is their ability to learn these filters/characteristics. Given the abovel we propose the use of a CNN to classify the data coming from the multisensorial helmet.

The proposed CNN’s operation is illustrated in Figure 8. The CNN consists of segmenting groups of pixels close to the input image and mathematically operating against a small matrix called a kernel. However, the part of the image is replaced with the input vector of size 5, where a re-shape is made to obtain a vector of 5×1. Therefore, the kernel proposed in the current CNN is of size 1, and moves from 1×1 pixel, in our case it would be different dimensions. With that size it manages to visualize all the input neurons and thus it can generate a new output matrix; a matrix that will be our new layer of hidden neurons.

A CNN can contain the spatial and temporal dependency characteristics in an image by applying relevant filters, the same applies to a data set that has been re-organized. The proposed architecture is an input layer for the transformed vector with size 5×1×1 with two hidden convolutional layers for two-dimensional data (Conv2D) and ReLU activation functions with a total of 64 and 32 neurons respectively. Finally a layer with 12 output neurons with SoftMax activation function for multiclass classification.

A classical model of Adam was proposed and trained with 500 epochs, the parameters were the same for the static NN and CNN to have an objective margin with respect to their evaluation. The following are the results on the AI models used for their implementation in the multisensorial helmet. Table 11 shows the evaluation for CNN where an overall accuracy of 92.05% was achieved.

### 4.4. Results

As mentioned above, each model was evaluated with 20% of data for cross validation. The SVM presented a general accuracy of 68.51% which was the model with the lowest performance in cross-validation. Its behavior is compared with that of the rest of the analyzed classes in Figure 9. Therefore, the use of this model in the multisensory helmet has been discarded.

An average accuracy of 78.26% has been achieved by NB in all the classes, as shown in Figure 10. Its performance has been better in class 5 and class 11. Despite having a better result than SVM it has been discarded since there were models that had better performance.

Figure 11 and Figure 12 show the performance of static NN and CNN respectively. In Figure 8 it can be observed that there is not a significant difference in the performance of NB, which had an accuracy of 78.56%. On the contrary, CNN, which allows for the implicit extraction of characteristics and for maintaining the relationships between the information regarding the dataset, had a considerably better result, with an accuracy of 92.05%. Our innovation comes on the proposed implementation of a CNN in a safety helmet as a proposal to reduce accidents and fractures in work areas, also through the use of technologies such as IoT for rapid synchronization of alarms that are sent to supervisors to take immediate action.

Given the above, CNN is the model that has been implemented in conjunction with the ThingsBoard platform. ThingsBoard and CNN work independently, creting an alarm system in a simulated environment that can serve as an higher security approach to a work environment. CNN is in charge of validating the information obtained from the platform, see Figure 13.

Previously it was mentioned that the creation of the deep models was through the “trial and error” approach, but the possible problem of overfiting should not be left aside, that is why Table 12 shows the results for the CNN in 10-Folds that shows the average performance from an objective point of view of the models.

In the next section, the conclusions drawn from the conducted research are described, and the contributions of this work to the state of the art are highlighted.

## 5. Conclusions and Discussion

Our work has a history of electronic development in which the use of a multisensory helmet was established. Through a conditional model of input–output rules, we tried to detect the different situations to which a worker was subjected. However, the input–output techniques presented false positives and false negatives with 60% accuracy in the best of cases, which is why after several stages, it was decided to implement AI in the helmet. The 60% that was described a moment ago is due to the combination of different circumstances, that is to say, the correlation that exists on the independent characteristics, is for that reason, that through techniques that find linear and nonlinear relations we decided to innovate in the present work. Since it is necessary to find the patterns that determine a particular action, for them there are the techniques of deep learning as our work presents

The comparison between different models of AI has been made in this research. Our innovation comes from the moment of using a CNN that in the literature has been used to analyze images or videos in intelligent helmets with the aim of saving lives. However, we proposed a multisensory approach to real-time feature analysis. Through the transmission of data through specialized IoT devices, a smart helmet has been designed to monitor the conditions in a working environment. The application areas of this proposal are industrial and agricultural sectors and any other sector that involves risk for the workers. Thanks to the helmet, different injuries can be avoided, and in case an accident occurs the damage caused to the worker is lessened through prompt attention or detection.

It is possible to observe in Figure 9, Figure 10 and Figure 11 that the MSA presented many false positives on majority classes in sample size, and even false positives of repeated classes (class 6) on more than five different classifications. NB and NN had a better performance in minority classes, however, there are three different classifications in false positives in classes such as 11, 9, 7, 2, 1 and 0. The NN has a strong resolution where the classes mentioned above still present false positives but with a decrease to 2 wrong classes in almost all cases.

### 5.1. Limitations

The work has different limitations. It is well known that artificial intelligence has the ability to find patterns that can hardly be found in linear analysis models. However, as stated in [73] risk analyses are not yet common in project-oriented industries. A problem with current risk analysis procedures is that procedures that are simple enough to be used by normal project staff are too simplistic to capture the subtlety of risk situations. Those that are complex enough to capture the essence and subtlety of risk situations are so complex that they require an expert to operate them. That is why the combination of possible risk situations can be counterproductive in the industrial area, an area that should be analyzed in more detail, with the following consequences:False positives would result in economic losses that would eventually affect the services and production areas involved, since the medical service and will be attending to situations that were not risky, the industry part will have to make production stops every time a false positive is found.On the other hand, false negatives are even more dangerous because the misinterpretation of data due to the complexity that can cause the unbalance of classes with less data set would result in losses not only economic but also of human personnel due to situations that were not attended to in the indicated time.

Our system has limitations on the amount of data that can be processed due to the microcontroller and the data that the model supports through the ESP32 module. That is why other techniques can be adopted, as will be seen in the next part of future work.

### 5.2. Future Work Opportunities

The use of paradigms such as edge computing or fog computing for the processing of many data as would be the integration of images or video would be the viable option to allow a transmission of information in real time, avoiding problems of saturation by the microcontroller. Several state-of-the-art researches have proposed smart helmets, among them is the US6798392B2 patent [74], which integrates a global location system, an environmental interaction sensor, a mobile communications network device, a small display panel, a microphone and a speaker. The helmet knows the location of the user and their interaction with the environment. The helmet can provide data to a user, monitor the actions of the user and conditions. This work is quite interesting since it offers device–user interaction. On the contrary, the advantage of our proposal is that it strives towards the autonomy of the system where decisions are made by the convolutional method.

Furthermore, the US9389677B2 patent [75], which is a smart helmet that includes a camera, a communications subsystem and a control subsystem. The control subsystem processes the video data from the camera, and the communications subsystem transmits this video data from the smart helmet to a target device. This work can be taken as a reference for a future sensor integration, since in our proposal it would be possible to integrate a camera that can process data through Deep CNN, for example thermal radiation data or even data regarding those who are infected with COVID-19.

Furthermore, the US registered patent, US20150130945A1 [76], in which a smart helmet is proposed that includes a helmet shell, a visor and a projector mounted on the helmet shell. The projector is configured so that content can be selected for display on the visor. The visor is rotatably attached to the helmet shell, and is configured to expose or cover the passage. The hull of the helmet defines an internal cavity and a passage that communicates with the internal cavity. The internal cavity is designed to receive the head of a user. This proposal’s focus is directed at the ergonomic part for the user, in addition to having navigation systems and control modules. This research is comparable to our proposal.

Moreover, in 2013 a helmet was proposed by Rasli Mohd Khairul Afiq Mohd et al. [77] for the prevention of accidents in which an FSR and a BLDC fan were used to detect the head of the driver and the speed of the motorcycle, respectively. A 315 MHz radio frequency module was used as a wireless link for communication between the transmitter circuit and the receiver circuit. PIC16F84a is a microcontroller for the control of the different components of the system. The motorcyclist could start the engine only when they had fastened their helmet. In comparison, our proposal communication takes place through IIoT for optimized decision making in case of accidents.

With reference to smart helmets connected to IoT, in 2016 [78], Sreenithy Chandran et al. presented a design whose objective is to provide a channel and a device for monitoring and reporting accidents. Sensors, a Wi-Fi enabled processor, and cloud computing infrastructures were used to build the system. The accident detection system communicates the accelerometer values to the processor that continuously monitors erratic variations. When an accident occurs, details about the accident are sent to emergency contacts using a cloud-based service. The location of the vehicle is obtained using the global positioning system. This work has a close relationship with the one proposed by us where there is optimized communication to reduce the consequences of accidents, the approach is different since we propose it for a work environment that can later be adapted for a case focused on vehicle safety, mainly on motorcycles.

## Figures and Tables

**Figure 1 sensors-20-06241-f001:**
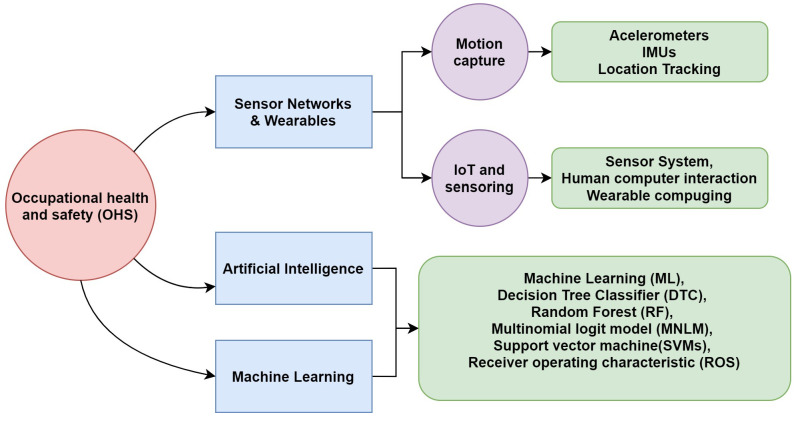
A block diagram of the devices.

**Figure 2 sensors-20-06241-f002:**
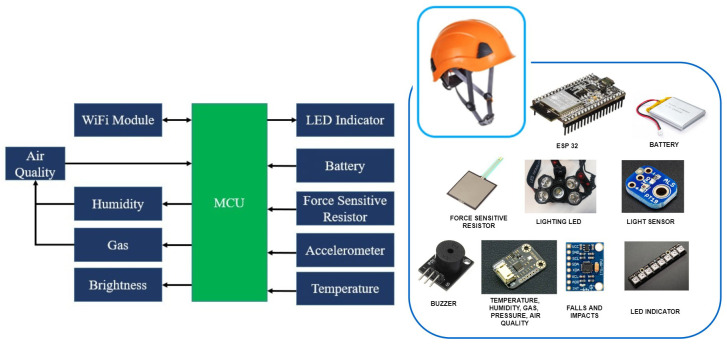
A block diagram of the devices.

**Figure 3 sensors-20-06241-f003:**
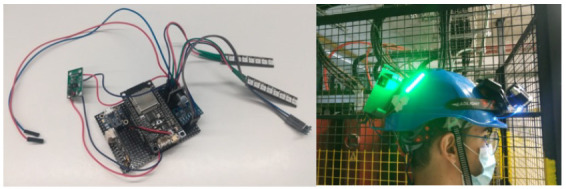
The electronic system of the helmet.

**Figure 4 sensors-20-06241-f004:**
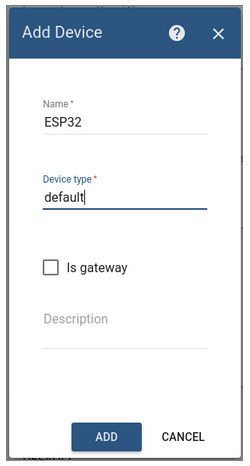
Setting up the ThingsBoard platform to operate according to the information received from ESP32, IoT module added to ThingsBoard and Multi-sensorial configuration.

**Figure 5 sensors-20-06241-f005:**
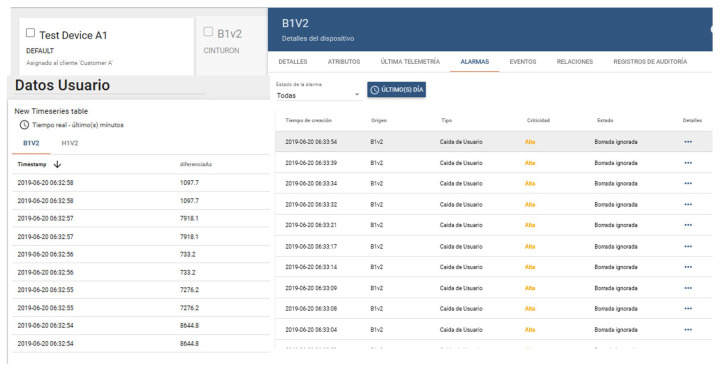
Alarm configuration on ThingsBoard, Block alarm creation method and Connecting alarms with sensors.

**Figure 6 sensors-20-06241-f006:**
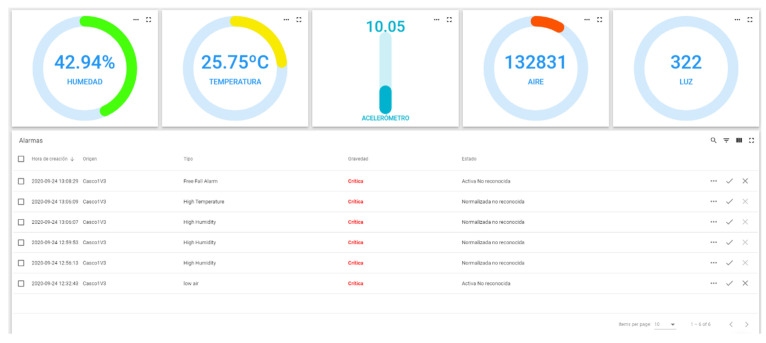
Final configuration of ThingsBoard platform to be validated through an intelligent algorithm.

**Figure 7 sensors-20-06241-f007:**
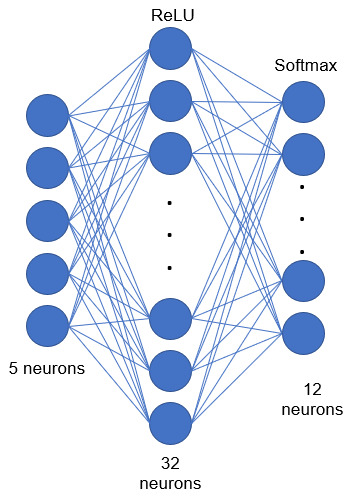
Proposed architecture, static neural network.

**Figure 8 sensors-20-06241-f008:**
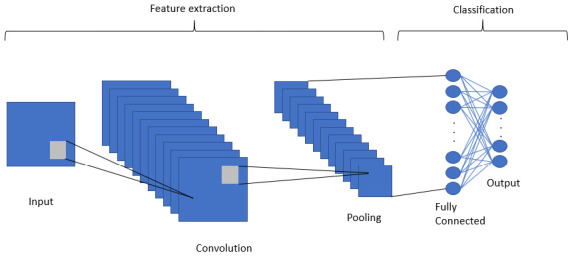
Deep convolutional neural network operation.

**Figure 9 sensors-20-06241-f009:**
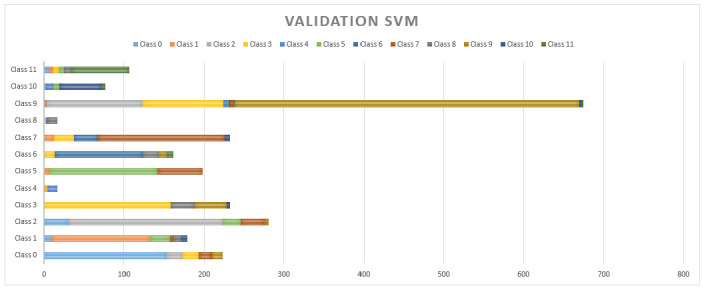
Cross-validation results with 20% for the SVM.

**Figure 10 sensors-20-06241-f010:**
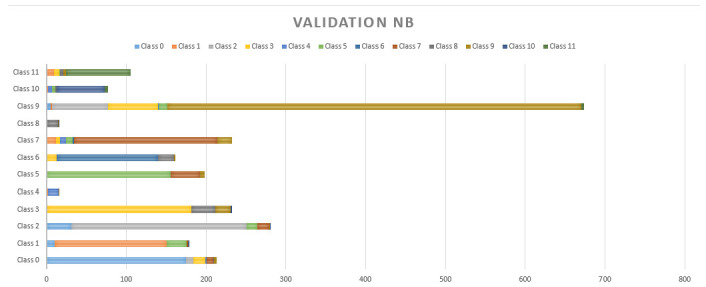
Cross-validation results with 20% for the NB.

**Figure 11 sensors-20-06241-f011:**
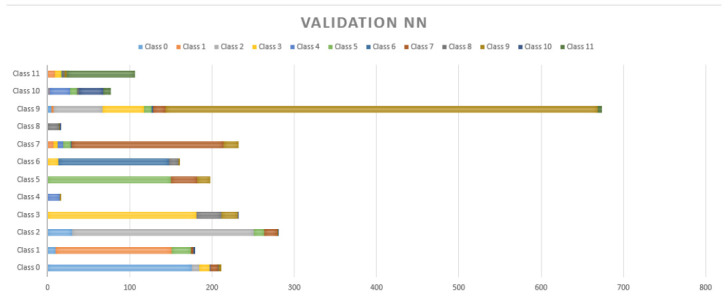
Cross-validation results with 20% for the NN.

**Figure 12 sensors-20-06241-f012:**
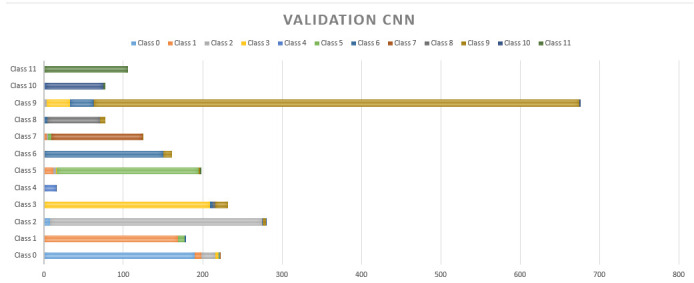
Cross-validation results with 20% for the CNN.

**Figure 13 sensors-20-06241-f013:**
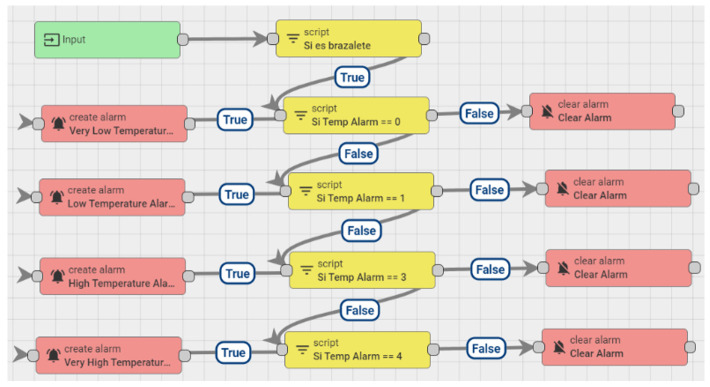
System of alarm rules established in ThingsBoard.

**Table 1 sensors-20-06241-t001:** OSH-related proposals.

Bibliography	Keywords	Novelty of the Proposal
Vaughn Jr, Rayford B. et al. (2002)	Security-engineering, Risk assessment	The state of security-engineering practices by three information security practitioners with different perspectives.
Choudhry, R. M., and Fang, D. (2008)		This work discusses empirical research aimed at why construction workers engage in unsafe behavior.
Niu, Yuhan, et al. (2019)		This research seeks to develop a smart construction object enabled OHS management system.
Champoux, D., and Brun, J. P. (2003)	Occupational health and safety (OHS), Construction safety, Artificial Inteligence (AI)	This exploratory study based on telephone interviews with the owner-managers of small manufacturing enterprises gives an overview of the most characteristic OHS representations and practices in small firms.
Podgorski, Daniel, et al. (2017)		A proposed framework based on a new paradigm of OSH risk management consisting of real-time risk assessment and risk level detection of every worker individually.
Barata, Joao et al. (2019)		Viable System Model (VSM) to design smart products that adhere to the organization strategy in disruptive transformations
Sun, Shengjing, et al. (2020)		A unified architecture to support the integration of different enabling technologies
Hasle, P., and Limborg, H. J. (2006)	Occupational health and safety, Accident Prevention	The scientific literature regarding preventive occupational health and safety activities in small enterprise.
Hasle, P., et al. (2011)	Occupational health and safety, Accident prevention	The investigation applied qualitative methods and theoretical approaches to CSR, small and medium-sized enterprises (SMEs), and occupational health and safety.
Abdelhamid, T. S., Everett, J. G. (2000)	Occupational safety, Construction safety, Accidents prevention	Accident root causes tracing model (ARCTM) tailored to the needs of the construction industry.
Chi, S., Han, S. (2013)		This study incorporates the systems theory into Heinrich’s domino theory to explore the interrelationships of risks and break the chain of accident causation.
Cambraia, F. B., et al. (2010)	Incident reporting systems, Safety management	Guidelines for identifying, analyzing and disseminating information on near misses in construction sites.
Chevalier, Yannick, et al. (2004)	Network security, Cryptographic protocols	High level protocol specification language for the modelling of security-sensitive cryptographic protocols.

**Table 2 sensors-20-06241-t002:** Proposals related to sensor networks.

Bibliography	Keywords	Novelty of the Proposal
Zhou, Yinghui, et al. (2012)	Internet of Things, Wearable Computing, Robot sensing systems, Acceleration Feature analysis, Human-computer interaction	Wearable device based on a tri-axis accelerometer, which can detect acceleration change of human body based on the position of the device being set.
Zhu, C., and Sheng, W. (2009)		A human daily activity recognition method by fusing the data from two wearable inertial sensors attached on one foot and the waist of the subject.
Lindeman, Robert W., et al. (2006)		A development history of a wearable, scalable vibrotactile stimulus delivery system.
Kim, Sung Hun, et al. (2018)		Experiments were performed in which the sensing data were classified whether the safety helmet was being worn properly, not worn, or worn improperly during construction workers’ activities.
Nithya, T., et al. (2018)	Head motion recognition, Hazardous gas, Temperature measurement, Sensor System, IMU, Electroencephealography (EEG)	Smart helmet able to detect hazardous events in the mining industry and design a mine safety system using wireless sensor networks.
Li, Ping, et al. (2014)		Smart Safety Helmet (SSH) in order to tack the head gestures and the brain activity of the worker to recognize anomalous behavior.
Fang, Y., et al. (2016)	Crane safety, Human errorReal-time, Crane motion capturing	A prototype system was developed based on the framework and deployed on a mobile crane.
Cao, Teng, et al. (2014)	Steady-state visual evoked potential (SSCEP), Brain-computer interfaces (BCIs)	Propose a method for the real-time evaluation of fatigue in SSVEP-based BCIs.

**Table 3 sensors-20-06241-t003:** Proposals related to safety environment and motion recognition.

Bibliography	Keywords	Novelty of the Proposal
Fernández-Muñiz, B., et al. (2012)	Safety climate, Employee perceptions, Safety performance	The current work aims to analyse the safety climate in diverse sectors, identify its dimensions, and propose to test a structural equation model that will help determine the antecedents and consequences of employees’ safety behaviour.
Glendon, A. I., Litherland, D. K. (2001)		A behavioral observation measure of safety performance and a road construction organization using a modified version of the safety climate questionnaire (SCQ).
Han, Y., and Song, Y. H. (2003)	IMU, Magnetometers, Gyroscopes, Accelerometer, Human motion	After introducing the concepts and functions of CM, this paper describes the popular monitoring methods and research status of CM on transformer, generator, and induction motor, respectively.
Godfrey, A. C. R. M. D. O. G., et al. (2008)		The underlying biomechanical elements necessary to understand and study human movement.
Ohtaki, Y., et al. (2001)		A new method is proposed to investigate kinematics and dynamics of locomotion without any limitation of laboratory conditions.
Zampella, Francisco, et al. (2012)		The usage of the Unscented Kalman Filter (UKF) as the integration algorithm for the inertial measurements.
Cheng, T., and Teizer, J. (2013)	Body sensor network (BSN), Vision Algorithms, Augmented reality (AR), Virtual Reality (VR), Location tracking	A novel framework is presented that explains the method of streaming data from real-time positioning sensors to a real-time data visualization platform.
Bleser, Gabriele, et al. (2015)		Assistance system based in the last advances in hardware, software and system level.
Fitton, Daniel, et al. (2008)		Investigation into how physical objects augmented with sensing and communication technologies can measure use in order to enable new pay-per-use payment models for equipment hire.
Yu, H., et al. (2007)	Measuring vigilance, Sensor network, Intelligent sensors	Signal transform method, Common Spatial Pattern, to process the EEG data.
Qiang, Cheng, et al. (2009)		A cost effective ZigBee-based wireless mine supervising system

**Table 4 sensors-20-06241-t004:** Proposals related to Smart manufacturing and Machine Learning.

Bibliography	Keywords	Novelty of the Proposal
Lee, Jay, et al. (2018)	Artificial Inteligent, Smart manufacturing, Failt diagnosis	State of AI technologies and the eco-system required to harness the power of AI in industrial applications.
Henley, E. J., and Kumamoto, H. (1985)		Provides a quantitative treatment of the optimal design of safety systems focusing on information links (human and computer), sensors, and control systems.
Li, Bo-hu, et al. (2017)		Based on research into the applications of artificial intelligence (AI) technology in the manufacturing industry in recent years.
Xiaoli, X. et al. (2011)		A presentation of Intelligent internet of things for equipment maintenance (IITEM) which we can make intelligent processing of device information.
Varian, Hal. (2018)		Summary of some of the forces at work and to describe some possible areas for future research.
Wahab, L., and Jiang, H. (2019)	Machine Learning (ML), Decision Tree Classifier (DTC), Random Forest (RF), Multinomial logic model (MNLM), Support vector machine (SVMs), Receiver operating characteristic (ROS)	Traffic crash analysis using machine learning techniques.
Azar, A. T., et al. (2014)		A random forest classifier (RFC) approach is proposed to diagnose lymph diseases.
Belgiu, M., and Drăguţ, L. (2016)		This review has revealed that RF classifier can successfully handle high data dimensionality and multicolinearity, being both fast and insensitive to overfitting.
Khalilia, M., et al.		Method for predicting disease risk of individuals using random forest.
Jedari, E., et al. (2015)		Machine learning approaches including k-nearest neighbor (k-NN), a rules-based classifier (JRip), and random forest have been investigated to estimate the indoor location of a user or an object using RSSI based fingerprinting method.
Iranitalab, A., and Khattak, A. (2017)		This paper had three main objectives: comparison of the performance of four statistical and machine learning methods including Multinomial Logit (MNL), Nearest Neighbor Classification (NNC), Support Vector Machines (SVM) and Random Forests (RF), in predicting traffic crash severity.
Pal, M. (2005)		To present the results obtained with the random forest classifier and to compare its performance with the support vector machines (SVMs) in terms of classification accuracy, training time and user defined parameters.
Rodriguez-Galiano, V. F., et al. (2012)		The performance of the RF classifier for land cover classification of a complex area is explored.
Yogameena, B., et al. (2019)	Complex software system, Mixture models, Convolutional neural networks	Intelligent video surveillance system for automatically detecting the motorcyclists with and without safety helmets.
Cockburn, D. (1996)		The benefit of taking a holistic perspective to developing complex software systems.

**Table 5 sensors-20-06241-t005:** Identification of common risk situations in the worker’s environment.

Risk Factors	Associated Hazards
Lack of Adequate Lighting	- The inability of the worker to see their environment clearly leads to accidental hits, slips, trips and falls. - The worker is unaware of the events occurring in their environment.
Temperature	- Extreme temperature changes leading to a heat stroke
Air Quality	- Harmful air in the environment
Operator Movement	- Slips, trips and falls - Blows to the worker’s head

**Table 6 sensors-20-06241-t006:** Identification of electronic components for the prevention of risks in the worker’s environment.

Risk Factors	Solution
Lack of Adequate Lighting	- Implementation of a brightness sensor in the helmet - Inclusion of torches as one of the tools of worker
Temperature	- Implementation of temperature sensors in the devices of the worker or environment
Air Quality	- Moisture and gas sensors.
Operator Movement	- The use of wearable devices with accelerometers capable of detecting falls. - Integration of sensitive force resistors in the helmet of the operator.

**Table 7 sensors-20-06241-t007:** Parameters for which data are collected.

1. Brightness
2. Variation in *X*, *Y* and *Z* axis
3. Force Sensitive Resistor
4. Temperature, Humidity, Pressure
5. Air quality

**Table 8 sensors-20-06241-t008:** Confusion matrix SVM.

Predicted Class (Vertical)/True Class (Horizontal)	Class 0	Class 1	Class 2	Class 3	Class 4	Class 5	Class 6	Class 7	Class 8	Class 9	Class 10	Class 11
**Class 0**	153	10	31	0	0	0	0	0	0	1	1	6
**Class 1**	0	122	0	0	1	6	0	12	0	2	1	5
**Class 2**	20	0	192	0	0	0	0	0	0	120	0	0
**Class 3**	20	0	0	158	3	0	13	25	2	101	0	7
**Class 4**	1	0	0	0	12	0	0	0	3	5	9	0
**Class 5**	0	25	23	0	0	135	0	0	0	2	8	7
**Class 6**	0	0	0	0	0	0	110	30	0	1	0	0
**Class 7**	15	5	30	0	0	57	0	159	0	5	0	0
**Class 8**	1	9	0	30	0	0	20	0	11	1	0	9
**Class 9**	13	0	5	40	0	0	10	0	0	432	0	0
**Class 10**	0	8	0	4	0	0	0	6	0	3	53	0
**Class 11**	0	0	0	0	0	0	8	0	0	2	5	72

**Table 9 sensors-20-06241-t009:** Confusion matrix NB.

Predicted Class (Vertical)/True Class (Horizontal)	Class 0	Class 1	Class 2	Class 3	Class 4	Class 5	Class 6	Class 7	Class 8	Class 9	Class 10	Class 11
**Class 0**	174	10	31	0	1	0	0	0	0	5	1	0
**Class 1**	0	140	0	0	1	0	0	12	0	2	1	9
**Class 2**	10	0	220	0	0	0	0	0	0	70	0	0
**Class 3**	15	0	0	181	0	0	13	5	1	62	0	7
**Class 4**	1	0	0	0	13	0	0	7	0	1	5	0
**Class 5**	0	25	13	0	0	155	0	9	0	11	4	1
**Class 6**	0	0	0	0	0	0	126	1	0	0	0	0
**Class 7**	9	2	15	0	0	37	0	180	0	0	0	0
**Class 8**	1	1	0	30	0	0	20	0	13	0	2	4
**Class 9**	3	0	1	20	1	6	2	18	1	520	0	3
**Class 10**	0	1	1	1	0	0	0	0	1	0	60	0
**Class 11**	0	0	0	0	0	0	0	0	0	3	4	82

**Table 10 sensors-20-06241-t010:** Confusion matrix NN.

Predicted Class (Vertical)/True Class (Horizontal)	Class 0	Class 1	Class 2	Class 3	Class 4	Class 5	Class 6	Class 7	Class 8	Class 9	Class 10	Class 11
**Class 0**	175	10	30	0	0	0	0	0	0	5	1	0
**Class 1**	0	141	0	0	0	0	0	7	0	2	1	9
**Class 2**	9	0	221	0	0	0	0	0	0	60	0	0
**Class 3**	13	0	0	181	0	0	13	5	0	50	0	7
**Class 4**	1	0	0	0	15	0	0	7	0	0	25	0
**Class 5**	0	23	12	0	0	150	0	9	0	9	9	1
**Class 6**	0	0	0	0	0	0	134	1	0	2	0	0
**Class 7**	9	3	16	0	0	32	0	185	0	15	0	0
**Class 8**	1	1	0	30	0	0	12	0	14	0	2	4
**Class 9**	3	0	1	20	1	16	2	18	1	526	0	2
**Class 10**	0	1	1	1	0	0	0	0	1	0	30	0
**Class 11**	0	0	0	0	0	0	0	0	0	5	9	83

**Table 11 sensors-20-06241-t011:** Confusion matrix CNN.

Predicted Class (Vertical)/True Class (Horizontal)	Class 0	Class 1	Class 2	Class 3	Class 4	Class 5	Class 6	Class 7	Class 8	Class 9	Class 10	Class 11
**Class 0**	190	0	8	0	0	0	0	0	0	0	0	0
**Class 1**	8	169	0	0	0	12	0	3	0	0	0	0
**Class 2**	18	0	267	0	0	3	0	2	0	3	1	0
**Class 3**	4	0	0	209	0	1	0	0	0	30	0	0
**Class 4**	1	0	1	0	15	0	0	0	0	0	1	0
**Class 5**	1	8	0	0	0	179	0	4	0	0	0	0
**Class 6**	0	2	0	4	0	0	150	0	4	29	0	0
**Class 7**	1	0	0	0	0	0	0	115	0	0	0	0
**Class 8**	0	0	0	3	0	0	0	0	67	1	0	0
**Class 9**	0	0	4	16	0	2	11	1	6	612	0	0
**Class 10**	0	0	1	0	1	0	0	0	0	1	73	0
**Class 11**	0	0	0	0	0	1	0	0	0	0	2	106

**Table 12 sensors-20-06241-t012:** Ten-fold validation for CNN.

Ten Fold Cross-Validation Test Sets	Accuracy (%) (Automated Risk Situations Develop in This Research)
1	93.18
2	93.09
3	90.73
4	94.12
5	91.27
6	92.75
7	92.61
8	92.59
9	92.76
10	91.99
**Average Accuracy**	92.509

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
