# Peer review of "Smart Helmet 5.0 for Industrial Internet of Things Using Artificial Intelligence"

_sensors, 2020, doi:10.3390/s20216241_

Round 1

Reviewer 1 Report

I found the manuscript very interesting from the point of view of multisensors application. The prototype helmet presented is quite a good tools fot safety. However, before to be acceptable for publication the paper requires revision in the part concerning the artificial intelligence discussion. In my opinion the authors should explain how they dimension the various AI. For example the hidden layer of static NN why is formed by 32 neurons and why you use ReLU. Maybe you used any approach to best design or just it is a results of a "try and correct" approach or ... Please add details. Also the presentation of the results in terms of matrix confusion seems to be confusing. How many were elements for each class selected in training and in validation? The confusion matrix reports test or validation results (I assume that is it the validation, but then why proposes also graph, i.e. fig. 9 -?) Are you using the same elements for all methods? If not this is a problem for the comparison. For example reading the SVM confusion matrix seems that class 7 has 108 elements (and none is assigned to class 7), whereas in NB the elements are 88 and in NN the number is higher. In addition maybe the results should be proposed in graphs (fig. 9 and so on, if these figg are manteined) in % since there are too different in the number of elements for each class.

Typos:

There are some typos in the manuscript. Some regards the numbers (for example 1175,5 instead of 11755. Please check accurately 

Author Response

Dear Reviewer

Thank you very much for the advice and guidelines you have given me in the article.

I have made all the corrections, currently shown in blue. I hope that you have managed to achieve the objectives that were intended in the review

Kind regards

Reviewer 2 Report

Dear Authors,

This paper dealt with a smart helmet equipped with several sensors by processing the sensing data using AI-driven platform. The performance of some supervised learning methods such as CNN, NN, NB, and SVM were compared while learning the training data consisting of 11755 samples and 12 different scenarios. I think this study is about a practical approach, but it doesn’t seem that the proposed helmet system was tested in outdoor/indoor environments by reacting or avoiding the accidents. In the 4.4 results, there are only comparisons of the performance of supervised learning methods. However, many researchers already know that the CNN has better performance than others introduced in this study in many cases. It is not new thing. Thus, careful tests and analysis are required to demonstrate the strength of the smart helmet.

The followings are questions for this study.

1.What is the history of the Smart helmet 5.0 platform? what is the meaning of the development of the fifth version of the smart helmet?

  1. Line 6: one 'continuously' should be removed.
  2. The author should use not ',' but a period or 'and' at the end of the sentence. Ex) Line 8.
  3. Line 226: what is c in Eq.(1) ?
  4. Line 339: have a The mathematical -> have the mathematical
  5. Line 353: , this is finding a b and w such-> and this is finding b and w such that
  6. The image quality should be better: Do check (Figure 1. ~ Figure 13)
  7. The author said that "Through the transmission of data through specialized IoT devices, a smart helmet has been designed to monitor and control the conditions in a working environment". What is the control of the condition? In this study, there was no explanation for control. The authors only described the detection of 12 situations according to test data and the supervised learning methods.
  8. Line 341:  1175,5 -> 11755

Author Response

(The authors gave the same response as above.)

Reviewer 3 Report

Hi authors,

Attached, please find my comments.

Best Regards,

Author Response

(The authors gave the same response as above.)

Reviewer 4 Report

This paper presents a smart helmet development and a comparison of AI learning models for the detection of occupational risks. The authors results present a CNN accuracy of 92.05%. The creation of smart objects for Occupational Health and Safety is a popular line of research and is expected to become a standard in the future of many professions. Industry, agriculture, or construction (one of the most hazardous sectors of the economy) are examples of sectors that have an urgent need of new solutions.

However, in my opinion, there are some critical problems in the paper that require the authors attention:

  1. The benefit of adopting AI to “notify the employees and their supervisors of any anomalies and threats” is nor properly explained in the introduction. Why using AI if it is possible to use thresholds for each critical parameter and only one is sufficient to justify an alarm (e.g. air quality with dangerous gases present)?
  2. The flow of the paper is confusing and requires some changes
  3. The review with a chronologic order does not seem to be a good option
  4. Lacking conclusions, limitations, and future work.

The detailed review is presented as follows:

Title: I do not see a connection with “Industrial Internet of Things”. Many sectors of industry do not require a smart helmet. Conversely, some sectors like mining, have dangerous environmental conditions that need to be monitored continuously. Perhaps the title can be more specific about the integration of AI in smart helmets.

Abstract:

The sentence “Paradigms such as the (…) make it possible to generate PPE  models feasibly” is unclear.

The smart helmet seems to be a prototype, and the focus is on the comparison of AI approaches. If that is the case, it should be clearer in the abstract.

Impact is missing. Why is this research important for theory and practice.

Introduction

First paragraph lacks a reference. Please check other cases.

Some vague sentences require a revision. For example (but there are others, please revise carefully the logic of all sentences in the paper): “Different studies have been conducted, which indicate the need to implement increasingly innovative solutions for workers in high-risk areas”. Studies about what? Smart helmets?

The following sentence is incorrect: “All the researches that address the problem of occupational safety and health (OSH) are given in Table 1”. First, please avoid the term “all the researches” because it is vague, and obviously inconsistent. The list is interesting but is dated, and misses integrated models that aim to address holistic adoption of intelligent systems to OHS (monitoring the person and its environment). For example (but there are others):

Barata, João, and Paulo Rupino da Cunha. "The Viable Smart Product Model: Designing Products that Undergo Disruptive Transformations." Cybernetics and Systems 50.7 (2019): 629-655.

Podgorski, Daniel, et al. "Towards a conceptual framework of OSH risk management in smart working environments based on smart PPE, ambient intelligence and the Internet of Things technologies." International Journal of Occupational Safety and Ergonomics 23.1 (2017): 1-20.

Sun, Shengjing, et al. "Healthy Operator 4.0: A Human Cyber–Physical System Architecture for Smart Workplaces." Sensors 20.7 (2020).

Section 2

The chronologic order of the review does not seem to be the best option. The studies are interesting but could be grouped in similar topics.

The paper does not present the sources and how the literature review was conducted. Therefore, it is not possible to confirm if the tables are complete. It is suggested to present the studies in Tables 2-4 as examples and inspiring research to support authors work.

Titles can be more explicit (e.g. Proposals related AI and Machine Learning).

Figure 1 seems to be part of section 3

Perhaps rename related works to “Background”. Related works seem more suitable to a list of studies using AI in Smart PPE systems.

Section 3

Line 197. Please include reference to support the selections in Table 6.

Table 5 and Table 6 include the parameters selected by the authors, which is OK. However, it is not possible to generalize these parameters to all types of workplaces. For example, some critical elements are missing such as noise (very common in some industries) or PPE detection (if the PPE is not used, the effort is pointless). Therefore, it is recommended to clearly state to which type of workplaces the proposed parameters would be more valuable.

Figure 2 can be more interesting if the sensor types are included near each element.

Why equation 1 appears and what is the rationale for the formula?

Line 220 – 251 can be improved. It is a sequence of statements. Could be presented as a table.

Figure 3 – Could be interesting to include a picture with a human using the helmet

Section 3.2. needs to be more compact. The figures are illegible.

Section 4.

Please revise the English and make sentences more objective. For example, “different algorithms used in the state of the art”.

Please explain the reason for the 12 labels.

Figure 5 is a discussion section. Please discuss the applicability of using AI in a real environment, for example, risks of performance, and advantage comparing to other possible methods to warn the user.

There are many risks involved in using an AI approach in situations that may have dramatic consequences. Please elaborate on this topic and explain how other measures could be included, for example, safety thresholds in each parameter and using the proposed approach to anticipate complex situations where a combination of lower levels of the parameters can produce accidents.

It is not clear how the classification of hazards (cause-consequence) appears in the models.

Please add conclusions, limitations, and future work opportunities, separately.

The topic is very interesting, and the comparison is also useful for future research efforts. I hope these comments can assist the authors in the necessary improvements.

Author Response

(The authors gave the same response as above.)

Round 2

Reviewer 1 Report

None further comment from my side. The authors have satisfactory answered to my comments.

Author Response

Thank you for the review and your time

Reviewer 2 Report

NA

Author Response

Thank you for the review and your time

Reviewer 3 Report

NA

Author Response

Thank you for the review and your time

Reviewer 4 Report

The authors have made significant improvements in the paper.

"The conditions in the workers’ environment are monitored permanently by this prototype helmet and the extracted data are transmitted to a server for processing." perhaps change to "This paper presents a smart helmet prototype that monitors the conditions in the workers’ environment and performs a near real-time evaluation of risks" (?)

Introduction
First paragraph lacks a reference. Please check other cases.

Figure 1 seems to be part of section 3

Line 197. Please include reference to support the selections in Table 6.

A final english revision is suggested.

Author Response

Thank you very much for your time and review.

We have made all the revision suggestions, except the one of moving the image, since as it is quoted in the text, the image shows those technologies that have been argued in the different works cited within the state of the art.

Please do not hesitate to let us know if you have any further revisions